# Towards Reliable and Holistic Visual In-Context Learning Prompt Selection

**Wenxiao Wu**[1,2]  **Jing-Hao Xue**[3]  **Chengming Xu**[4✉]  **Chen Liu**[5]  **Xinwei Sun**[6]
**Changxin Gao**[1]  **Nong Sang**[1]  **Yanwei Fu**[6,2]

[1] Huazhong University of Science and Technology    [2] Shanghai Innovation Institute
[3] University College London    [4] Tencent Youtu Lab
[5] The Hong Kong University of Science and Technology    [6] Fudan University
{wenxiaowu, cgao, nsang}@hust.edu.cn  jinghao.xue@ucl.ac.uk
chengmingxu@tencent.com  cliudh@connect.ust.hk
{sunxinwei, yanweifu}@fudan.edu.cn

## Abstract

Visual In-Context Learning (VICL) has emerged as a prominent approach for adapting visual foundation models to novel tasks, by effectively exploiting contextual information embedded in in-context examples, which can be formulated as a global ranking problem of potential candidates. Current VICL methods, such as Partial2Global and VPR, are grounded in the similarity-priority assumption that images more visually similar to a query image serve as better in-context examples. This foundational assumption, while intuitive, lacks sufficient justification for its efficacy in selecting optimal in-context examples. Furthermore, Partial2Global constructs its global ranking from a series of randomly sampled pairwise preference predictions. Such a reliance on random sampling can lead to incomplete coverage and redundant samplings of comparisons, thus further adversely impacting the final global ranking. To address these issues, this paper introduces an enhanced variant of Partial2Global designed for reliable and holistic selection of in-context examples in VICL. Our proposed method, dubbed RH-Partial2Global, leverages a jackknife conformal prediction-guided strategy to construct reliable alternative sets and a covering design-based sampling approach to ensure comprehensive and uniform coverage of pairwise preferences. Extensive experiments demonstrate that RH-Partial2Global achieves excellent performance and outperforms Partial2Global across diverse visual tasks. The source code is available in https://github.com/Wu-Wenxiao/RH-Partial2Global.

## 1  Introduction

Inspired by the success of in-context learning in natural language processing, visual in-context learning (VICL) was born on demand as a promising paradigm for leveraging Visual Foundation Models (VFMs) in vision tasks. By conditioning VFMs on a few in-context examples, typically image-label pairs, VICL facilitates efficient adaptation to various downstream tasks, such as segmentation and detection [1, 2], image editing [3], cross-modal content reasoning [4] as well as low-level tasks[5].

A key challenge in VICL lies in selecting the most suitable in-context examples corresponding to a given query image to achieve optimal performance. Given that a set of candidate examples is typically available, recent works [2, 6] commonly formulate VICL as a global ranking problem. Partial2Global [6], a state-of-the-art method, tackles this by first training a list-wise partial ranker through meta-learning. This ranker is applied to randomly partitioned candidate sequences for partial rankings, from which a consistency-aware aggregator subsequently infers a global ranking.

---

✉Corresponding author.

While Partial2Global marks a significant step forward for VICL, two critical limitations remain unresolved: **(1)** ***The similarity-priority assumption.*** Both Partial2Global and VPR rely on a heuristic known as the similarity-priority assumption that the more visually similar a candidate image is to the query, the better it serves as an in-context example. Although Partial2Global questions the reliability of this assumption, it lacks a statistical foundation to support this critique and does not offer concrete alternatives. As a result, the assumption remains unchallenged in a rigorous or systematic way, and its influence on performance is left largely unexplored. **(2)** ***The strategy of random sampling.*** In constructing a global ranking from partial rankings, it is ideal to uniformly and exhaustively cover all pairwise preferences among candidates. However, the reliance of Partial2Global on random shuffle operations for partial predictions often fails to capture all inter-candidate relationships and may introduce redundant comparisons while neglecting informative ones, ultimately degrading ranking accuracy. Thus, a more principled sampling strategy is needed to achieve comprehensive coverage of candidate relationships while simultaneously minimizing redundancy and iteration overhead.

Towards **R**eliable and **H**olistic VICL Prompt Selection, we build upon the **Partial2Global** framework and introduce an enhanced approach named **RH-Partial2Global**. To address the two key limitations discussed earlier, RH-Partial2Global incorporates the following major improvements: (1) *Reliable Selection Strategy via Conformal Prediction.* To construct a more trustworthy alternative set of in-context examples, we propose a selection strategy based on conformal prediction. This method operates in a jackknife manner: for each training sample, it computes a consistency score that quantifies the alignment between the sample's quality when serving as an in-context prompt for other instances and its visual similarity to these prompted instances. By applying a threshold derived from a quantile of these scores at a predefined confidence level, we identify a subset of candidates exhibiting high reliability. This identified subset is then used to filter the initial similarity-driven candidate pool, resulting in a more robust and accurate selection of in-context examples. (2) *Holistic and Efficient Sampling Strategy via Covering Design.* To ensure uniform and comprehensive coverage of pairwise candidate preferences, we integrate a covering design into the local sampling process. This is operationalized by sequentially sampling from a randomly shuffled alternative set, guided by a precomputed optimal covering design. Please note that the principles of covering design inherently ensure exhaustive coverage of preference relationships, while its property of minimizing sampling iterations contributes to the uniformity of pairwise preference sampling.

We summarize our contributions as four-folds:

- We turn our attention to a necessary yet often neglected heuristic in VICL that images more visually similar to a query image serve as better in-context examples. For the first time, we provide statistical evidence demonstrating that this similarity-priority assumption is not sufficiently robust.

- We propose a jackknife conformal prediction-based example selection strategy to identify and preserve reliable samples in the alternative set, thereby reducing ranking complexity and yielding more accurate predictions.

- We incorporate a covering design-based sampling strategy within the consistency-aware ranking aggregator of Partial2Global, ensuring more holistic coverage of pairwise preferences and consequently leading to a more accurate global ranking.

- Extensive experiments substantiate that our proposed RH-Partial2Global can significantly surpass the state-of-the-art Partial2Global framework across diverse vision tasks.

## 2 Related work

**In-Context Learning.** The recent expansion in the scale of large-scale models has led to remarkable advances in their ability to perform in-context learning, a process where models adapt to new tasks by conditioning on a small set of examples rather than undergoing further training. Notably, both large language models (LLMs) [7] and their multi-modal successors [8] have demonstrated this capability. For instance, Pan et al. [9] leveraged in-context learning to construct symbolic representations that facilitate logical reasoning, while Zhang et al. [10] applied similar techniques to update factual content within LLMs. Parallel developments have emerged in the field of computer vision. Approaches such as MAE-VQGAN [1] and Painter [5] have shown that vision models can be trained to perform in-context learning by reconstructing masked regions in image grids composed of both support and query images. Building on this, VPR [2] addressed the challenge of selecting optimal in-context samples, introducing a metric network based on contrastive learning and performance evaluation. Prompt-SelF [11] extended this line of work by incorporating both fine-grained

and coarse-grained visual similarities, and proposed an ensemble approach that aggregates predictions from multiple permutations of in-context grids at inference time. Partial2Global [6] proposed a list-wise ranker and a globally consistent ranking aggregator aiming for global optimal in-context prompts. Our work aims to find better in-context example, but different from previous works, we break the assumption that similarity between images can safely lead to better in-context examples. Other than that, we focus on leveraging conformal prediction and covering design to build reliable and holistic in-context example selection process.

**Conformal Prediction.** Conformal prediction (CP) [12, 13, 14] is a distribution-free and model-agnostic methodology that generates prediction sets with theoretically guaranteed coverage probabilities. Unlike traditional point predictions, CP quantifies uncertainty by producing a set of plausible outcomes rather than a single value. This ensures that the true outcome (e.g., label, value, or element) is contained within the prediction set with a user-specified confidence level. Conformal prediction has garnered significant attention in various tasks that require rigorous uncertainty estimation, such as multi-class prediction [15], benchmarking of LLMs [16], and robotic trajectory prediction [17]. CP methodologies can be broadly categorized into several types: full CP, split CP (or inductive CP) [18], transductive approaches like jackknife CP [19, 20] and CP with cross-validation [21, 22], and conformal risk control [23, 24]. Among these variants, jackknife CP is known for its tendency towards conservative predictions, rendering it particularly suitable for applications demanding high reliability and robustness. In this paper, we leverage jackknife CP to refine the construction of alternative sets, aiming to retain highly reliable samples and thereby alleviate the difficulty of subsequent ranking tasks.

**Covering Design.** A covering design [25] is a fundamental combinatorial structure that addresses the problem of systematically covering all possible subsets of a fixed size within a larger set. Formally, a $(K, k, t)$ covering design is a collection of $k$-element subsets (called blocks) from a $K$-element set, such that every $t$-element subset of the $K$-set is contained in at least one block. A primary objective in covering designs is to find the minimum number of blocks, $C(K, k, t)$, satisfying the covering condition, highlighting their efficiency in ensuring comprehensive interaction coverage. In our work, motivated by the limitations of random sampling and the inherent need for complete and uniform coverage of pairwise preferences (i.e., $t = 2$) among candidate examples, we replace the original random sampling with a covering design-based strategy, thereby facilitating a more systematic, comprehensive, and balanced sampling of these pairwise relationships.

## 3 Methodology

**Preliminary of Partial2Global**: Adhering to the similarity-priority heuristic prevalent in methods like Visual Prompt Retrieval (VPR) [2] that images more visually similar to a query $x_q$ serve as better in-context examples, Partial2Global [6] constructs an alternative set $\mathcal{Y}_q$ of size $K$ from training set $\mathcal{X}_{trn} = \{x_i^{trn}\}_{i=1}^{M+1}$ for each query sample $x_q$ in test set $\mathcal{X}_{test}$. As for the training stage, Partial2Global proposes a transformer-based list-wise ranker $\phi_k$ of length $k$ for local ranking and trains it in a meta-learning manner. When it comes to the evaluation stage, Partial2Global systematically infers global rankings from local predictions. This process begins by first constructing an observation pool comprising $N_p$ randomly shuffled variants of the alternative set $\mathcal{Y}_q$. For each such permuted variant, the $K$ candidates are divided into into $\lceil \frac{K}{k} \rceil$ non-overlapping sub-sequences of length $k$. Each sub-sequence is then ranked using $\phi_k$ and the resulting pairwise preferences are aggregated into a preference vector $S^i$, which entries encode dominance relationships $(1, -1, \text{or } 0)$. Each vector $S^i$ is then converted into a pairwise indication set $E^i$, which explicitly lists all candidate pairs with non-neutral preference relationships derived from the ranker's predictions. The collection of these indication sets and preference vectors forms the basis for deriving a global ranking score vector $r$. The derivation of $r$ is then reformulated as a least squares problem using transformation matrix $D^i$, which encodes pairwise comparisons from $E^i$, leading to $\min_r \sum_{i=1}^{N_p} \frac{1}{2N_p} ||D^i r - S^i||_2^2$. Compared to naive ranking, this aggregation approach enhances both effectiveness through preference modeling and efficiency by mitigating sequential dependencies.

**Overview.** To advance VICL via superior in-context prompt selection, we propose two corresponding enhancements that address two key limitations in Partial2Global. First, to mitigate the suboptimal criteria used for constructing initial alternative sets, we rely on the theoretical foundation of jackknife conformal prediction (Sec. 3.1) to identify a prompt set consisting of high-confidence, reliable candidates. With this reliable set, we refine the initial alternative set retaining only those

Table 1: Experimental results of Spearman's rank correlation test for each fold. The number and proportion of samples with statistically significant monotonic associations ($p < 0.05$) and the average correlation ($\bar{\rho}$) are reported.

| | Fold-0 | Fold-1 | Fold-2 | Fold-3 |
|---|---|---|---|---|
| #($p < 0.05$) | 1786/2279 (78.37%) | 2906/3309 (87.82%) | 4152/5030 (82.54%) | 1584/1986 (79.75%) |
| $\bar{\rho}$ | 0.0548 | 0.0315 | 0.0345 | 0.0500 |

candidates present in their intersection, thereby the reliability of the inputs to the global ranking process. Once this high-confidence alternative set is established, local rankings are typically generated through random shuffling of candidates prior to aggregation. However, this process can lead to incomplete and redundant coverage of all pairwise relationships, potentially resulting in performance instability. To this end, we further introduce the use of covering designs (Sec. 3.2) to guide a more structured and comprehensive sampling strategy, ultimately leading to more stable and accurate global ranking performance.

### 3.1 Conformal prediction-guided strategy for reliable candidate selection

This section details the construction of a reliable alternative set $\mathcal{Y}_\alpha$ with a user-specified confidence level $\alpha$ for query images, guided by the theory of conformal prediction with jackknife.

**Motivation.** To examine the validity of the similarity-priority assumption, which posits that images more visually similar to a query image serve as better in-context examples, we conduct an experimental hypothesis test based on the training set $\mathcal{X}_{trn}^p$ from Pascal-$5^i$ [26] dataset for the segmentation task. For each sample $x_i$ in $\mathcal{X}_{trn}^p$ serving as the query, the remaining examples in $\mathcal{X}_{trn}^p$ are used as potential in-context prompts. We then derive two sequences for these potential examples: (1) their IoU scores when used as prompts for $x_i$, and (2) their visual similarity scores to $x_i$. To assess the association between the two sequences, we calculate the Spearman's rank correlation coefficient $\rho \in [-1, 1]$ and employ the Spearman's rank correlation test to obtain the p-value $p \in [0, 1]$. Here, a high Spearman correlation coefficient $\rho$ indicates observations have similar rankings in terms of both their IoU scores and their visual similarities, while a low $\rho$ suggests dissimilar rankings. Since the null hypothesis $H_0$ of the Spearman's rank correlation test posits no monotonic association between the two variables, a p-value below the significance level (e.g., 0.05) indicates rejection of $H_0$, thereby suggesting a statistically significant monotonic relationship. We conduct this procedure across all folds of $\mathcal{X}_{trn}^p$, calculating both the average correlation $\bar{\rho}$ and the number and proportion of query samples for which $p < 0.05$. The experimental results are summarized in Table 1.

**Analysis.** The experimental results presented in the second row of Table 1 consistently indicate across all folds that there indeed exists a statistically significant monotonic relationship between the quality of in-context examples (e.g., IoU in segmentation tasks) and visual similarities. However, the average Spearman correlation coefficients $\bar{\rho}$ presented in the third row of Table 1 are notably low. This suggests that while a general statistical association is frequently present, the strength of this monotonic relationship is often weak. These observations underscore the need for a more robust criterion than pure similarity alone. Such a criterion should selectively identify and retain genuinely reliable candidate examples from an initial alternative set $\mathcal{Y}_q$, which was broadly guided by the similarity-priority assumption, thereby refining the selection process. Hence our algorithm as follows.

**Algorithm.** We begin by directly using the original training set $\mathcal{X}_{trn} = \{x_i^{trn}\}_{i=1}^{M+1}$ as the "training set" for conformal prediction. Please note that we do not need $\mathcal{X}_{trn}$ to train a predictive model, because our conformity function $f(\cdot)$ will be just a predefined metric (e.g., the negative KL Divergence or the Spearman correlation) that quantifies the consistency between the quality of in-context examples $\mathcal{Q}$ and their visual similarities $\mathcal{S}$, as detailed below from Eq.(1) to Eq.(3).

Let $\mathcal{F}$ be a pretrained inpainting model. Mathematically, when a sample $x_i^{trn}$ is used as an in-context prompt, the VICL process for any query results in an output $\mathcal{F}(\cdot, x_i^{trn})$. Given a function q, which typically evaluates the performance of $\mathcal{F}(\cdot, x_i^{trn})$ with respect to $x_i^{trn}$, the quality of this VICL process can be denoted as $q(\mathcal{F}(\cdot, x_i^{trn}), x_i^{trn})$. Consequently, for each $x_i^{trn} \in \mathcal{X}_{trn}$, we assess its quality as a prompt by applying it as the prompt to all other samples $x_j^{trn}$ in $\mathcal{X}_{trn} \setminus x_i^{trn}$. This yields a set of $M$ quality scores:

$$\mathcal{Q}(x_i^{trn}) = \{q(\mathcal{F}(x_j^{trn}, x_i^{trn}), x_i^{trn})\}_{j=1, j\neq i}^{M+1}. \tag{1}$$

---
**Algorithm 1** Jackknife Conformal Prediction-guided Candidate Selection
---
**Require:** Constructed training set $\mathcal{X}_{trn}$, query sample $x_q$, alternative set size $K$, pretrained inpainting model $\mathcal{F}$, conformity function $f$, confidence level $\alpha$

 1: Initial the conformity score set $\mathcal{V} := \{-\infty\}$
 2: **for** $x_i^{trn}$ in $\mathcal{X}_{trn}$ **do**
 3:     Initial quality set $\mathcal{Q}(x_i^{trn}) := \emptyset$ and similarity set $\mathcal{S}(x_i^{trn}) := \emptyset$
 4:     **for** $x_j^{trn}$ in $\mathcal{X}_{trn} \setminus x_i^{trn}$ **do**
 5:         Compute the quality score as as Eq.(1): $\mathfrak{q}(\mathcal{F}(x_j^{trn}, x_i^{trn}), x_i^{trn})$
 6:         Update quality set: $\mathcal{Q}(x_i^{trn}) = \mathcal{Q}(x_i^{trn}) \cup \mathfrak{q}(\mathcal{F}(x_j^{trn}, x_i^{trn}), x_i^{trn})$
 7:         Compute the similarity score as Eq.(2): $\mathfrak{s}(x_j^{trn}, x_i^{trn})$
 8:         Update similarity set: $\mathcal{S}(x_i^{trn}) = \mathcal{S}(x_i^{trn}) \cup \mathfrak{s}(x_j^{trn}, x_i^{trn})$
 9:     **end for**
10:     Compute the conformity score as Eq.(3): $\ell(x_i^{trn}) = f(\mathcal{Q}(x_i^{trn}), \mathcal{S}(x_i^{trn}))$
11:     Update conformity score set: $\mathcal{V} = \mathcal{V} \cup \ell(x_i^{trn})$
12:     Compute the quantile $q_{1-\alpha}(\mathcal{V})$ as Eq.(5)
13:     Construct the reliable set as Eq.(6): $\mathcal{Y}_\alpha = \{x_i^{trn} \in \mathcal{X}_{trn} : \ell(x_i^{trn}) > q_{1-\alpha}(\mathcal{V})\}$
14: **end for**
15: Initialize alternative set $\mathcal{Y}_q$ for $x_q$: $\mathcal{Y}_q = \text{top-K}_{\hat{x} \in \mathcal{X}_{trn}} \mathfrak{s}(x_q, \hat{x})$
16: Obtain the refined alternative set $\mathcal{Y}_q^*$ as Eq.(7): $\mathcal{Y}_q^* = \mathcal{Y}_\alpha \cap \mathcal{Y}_q$
17: **return** $\mathcal{Y}_q^*$.
---

Similarly, we define $\mathfrak{s}(\cdot, x_i^{trn})$ as a function measuring the visual similarity between the prompt $x_i^{trn}$ and any other query. For each $x_i^{trn}$, we compute its similarities to all other samples in $\mathcal{X}_{trn} \setminus x_i^{trn}$, forming a set of $M$ similarity scores:

$$\mathcal{S}(x_i^{trn}) = \{\mathfrak{s}(x_j^{trn}, x_i^{trn})\}_{j=1, j \neq i}^{M+1}. \tag{2}$$

Based on the above definitions, the conformity score $\ell(x_i^{trn})$ for each $x_i^{trn}$ is calculated by applying a function $f$ (e.g., the negative KL Divergence or the Spearman correlation) to its corresponding quality and similarity sets:

$$\ell(x_i^{trn}) = f(\mathcal{Q}(x_i^{trn}), \mathcal{S}(x_i^{trn})). \tag{3}$$

This procedure is repeated for every $x_i^{trn} \in \mathcal{X}_{trn}$, treating each in turn as the prompt whose conformity score is being computed, to constitute a jackknife procedure. Then, a set including all $M + 1$ conformity scores can be denoted by

$$\mathcal{V} = \{\ell(x_i^{trn})\}_{i=1}^{M+1} \cup \{-\infty\}. \tag{4}$$

Subsequently, the corresponding $(1 - \alpha)$-quantile $q_{1-\alpha}$, which serves as the reliability threshold, is determined from the empirical distribution of conformity scores in $\mathcal{V}$:

$$q_{1-\alpha}(\mathcal{V}) = \text{the } \lceil (1-\alpha)(M+2) \rceil\text{-th smallest } \ell. \tag{5}$$

Crucially, this quantile $q_{1-\alpha}(\mathcal{V})$ is derived solely from the training set $\mathcal{X}_{trn}$ and is thus independent of any specific query $x_q$ from the test set or its initial candidate set. Thus, we can define a global set of highly reliable candidate prompts $\mathcal{Y}_\alpha$ derived from $\mathcal{X}_{trn}$:

$$\mathcal{Y}_\alpha = \{x_i^{trn} \in \mathcal{X}_{trn} : \ell(x_i^{trn}) > q_{1-\alpha}(\mathcal{V})\}. \tag{6}$$

This set $\mathcal{Y}_\alpha$ comprises samples whose conformity scores are in the top $\alpha$ percentiles, indicating high reliability. We now describe how this globally reliable set $\mathcal{Y}_\alpha$ is used to refine a query-specific alternative set $\mathcal{Y}_q$. For a given query $x_q$, we first construct the initial alternative set $\mathcal{Y}_q$ based on the similarity-priority assumption: $\mathcal{Y}_q = \text{top-K}_{\hat{x} \in \mathcal{X}_{trn}} \mathfrak{s}(x_q, \hat{x})$. The selection of candidates in $\mathcal{Y}_q$ can be seen as a procedure of conformal prediction test. Our core idea is to test whether one element $x_j^q$ in $\mathcal{Y}_q$ can also satisfy $\ell(x_j^q) > q_{1-\alpha}(\mathcal{V})$, i.e., $x_q^{trn} \in \mathcal{Y}_q$. The entire process can be viewed as filtering a known $\mathcal{Y}_q$ to obtain the refined set $\mathcal{Y}_q^*$ using $\mathcal{Y}_\alpha$, which can be expressed as

$$\mathcal{Y}_q^* = \mathcal{Y}_\alpha \cap \mathcal{Y}_q = \{x_j^q : \ell(x_j^q) > q_{1-\alpha}(\mathcal{V}), x_j^q \in \mathcal{Y}_q\}. \tag{7}$$

**Remark.** In the procedure for constructing $\mathcal{Y}_\alpha$, our primary focus is on evaluating the intrinsic reliability of each $x_i^{trn}$ as a potential prompt, independent of any specific test query $x_q$. Consequently, while no guaranteed optimality for any individual test query $x_q$, this strategy aims to ensure that the selected reliable prompts offer overall satisfactory performance by virtue of their proven reliability on $\mathcal{X}_{trn}$. The whole procedure is outlined in Algorithm 1.

### 3.2 Covering design-based strategy for holistic sampling

In this section, we elaborate on how to achieve more comprehensive and balanced sampling, guided by covering design principles, to facilitate a more accurate global ranking.

**Motivation.** Identifying optimal candidates from the available pool is a critical step in the selection of in-context examples. Partial2Global [6] builds a pool of randomly shuffled variants of the alternative set, partitions each variant into multiple non-overlapping sub-sequences and employs a meta-ranker to perform local ranking predictions. Based on these local predictions, partial pairwise preferences are inferred and subsequently aggregated into a pairwise indication set, from which a global ranking is derived by solving a least squares problem. While this scheme yields reasonable results, it suffers from two primary limitations:

(1) **Incomplete Coverage:** Despite generating numerous randomly shuffled variants of $\mathcal{Y}_q$ into the observation pool, the random sampling process does not guarantee that all possible pairwise relationships between candidates are captured. Given a typically alternative set size $K = 50$ and a local ranker length $k = 5$ as in Partial2Global, ensuring that every candidate pair is compared within at least one $k$-length sub-sequence corresponds to constructing a $C(50, 5, 2)$ covering design. In fact, as indicated in Theorem 1, a $C(50, 5, 2)$ covering design requires a minimum of 130 distinct $k$-length sub-sequences to guarantee full pairwise coverage. While random sampling in Partial2Global (i.e., with 50 sub-sequences) might generate numerous sequences, it lacks this systematic guarantee and thus may fail to capture all essential pairwise relationships.

(2) **Non-Uniform Preference Weighting:** Although Partial2Global claims to account for contradicting predictions, thereby offering the potential to correct erroneous ones, it does not control for the frequency of repeated pairwise preferences. This can lead to certain locally derived pairwise preferences being overrepresented, thereby adversely influencing the global ranking.

**Theorem 1 (Schonheim Lower Bound [27])** *Considering a $(K, k, t)$ covering design $C(K, k, t)$, the Schonheim lower bound for such a covering design's size is*

$$C(K, k, t) \geq \lceil \frac{K}{k} \lceil \frac{K-1}{k-1} \ldots \lceil \frac{K-t+1}{k-t+1} \rceil \ldots \rceil \rceil. \tag{8}$$

Covering designs offer a principled approach to mitigate both of these limitations effectively. The inherent combinatorial structure of covering design guarantees exhaustive coverage of all pairwise relationships with an optimally minimal or near-minimal number of sub-sequences, while the associated lower bound promotes balanced and uniform sampling.

When it comes to implementation, the pre-computed optimal covering designs allows for an efficient strategy: one can simply generate a randomly shuffled variant of the alternative set $\mathcal{Y}_q$ of size $K'$ and sample $k$-length sequences from it according to the structure of a predefined optimal covering design $C^*(K', k, t)$. This structured sampling strategy incurs negligible additional computational overhead while providing a strong guarantee of comprehensive and balanced pairwise coverage.

## 4 Experiments

We give the setups, and evaluate our RH-Partial2Global on several visual tasks.

**Dataset.** Following VPR [2] and Partial2Global [6], we adopt three visual tasks: foreground segmentation, single object detection, and image colorization. For the segmentation task, We utilize the Pascal-$5^i$ [26] dataset, which comprises four different image splits. Performance is reported using the mean Intersection over Union (mIoU) for each split, along with the average mIoU across all four splits. The Pascal VOC 2012 dataset [28] is employed for the single object detection task. Consistent with MAE-VQGAN [1], our evaluation subset includes only images containing a single object with Pascal annotations, excluding trivial cases where an object occupies more than 50% of the image area. For the colorization task, we sample a test set from the validation set of ILSVRC2012 [29] to evaluate model performance, using Mean Squared Error (MSE) as the evaluation metric.

**Implementation details.** Given that our proposed RH-Partial2Global method does not require additional model, we fully adopt the settings of Partial2Global [6]. Specifically, we train meta-rankers with both lengths of 5 and 10 for foreground segmentation and single object detection, while meta-rankers of length 3 and 5 are utilized for the colorization task. Following VPR, all visual similarity scores in our experiments are computed using the vision encoder of CLIP [30], which was pre-trained using multimodal contrastive learning. Additionally, we employ DINOv2 [31] as the feature

Table 2: Comparison of our proposed RH-Partial2Global with some state-of-the-art VICL methods.

| Method | Ref. | Seg. (mIoU) ↑ | | | | | Det. (mIoU) ↑ | Color. (MSE) ↓ |
| | | Fold-0 | Fold-1 | Fold-2 | Fold-3 | Avg. | | |
|---|---|---|---|---|---|---|---|---|
| MAE-VQGAN [1] | NIPS'22 | 28.66 | 30.21 | 27.81 | 23.55 | 27.56 | 25.45 | 0.67 |
| UnsupPR [2] | NIPS'23 | 34.75 | 35.92 | 32.41 | 31.16 | 33.56 | 26.84 | 0.63 |
| SupPR [2] | NIPS'23 | 37.08 | 38.43 | 34.40 | 32.32 | 35.56 | 28.22 | 0.63 |
| Partial2Global [6] | NIPS'24 | 38.81 | 41.54 | 37.25 | 36.01 | 38.40 | 30.66 | 0.58 |
| RH-Partial2Global (Ours) | – | **39.25** | **42.15** | **38.06** | **36.60** | **39.02** | **30.94** | **0.56** |
| prompt-SelF [11] | arXiv'2023 | 42.48 | 43.34 | 39.76 | 38.50 | 41.02 | 29.83 | — |
| Partial2Global+voting [6] | NIPS'24 | 43.23 | 45.50 | 41.79 | 40.22 | 42.69 | 32.52 | — |
| RH-Partial2Global+voting | – | **43.53** | **45.88** | **41.99** | **40.90** | **43.08** | **33.28** | — |

Figure 1: Qualitative comparison between our proposed RH-Partial2Global and Partial2Global in the foreground segmentation task. For each comparison item, we display the image grid following the input order of MAE-VQGAN: the first row contains the in-context example alongside its corresponding label, while the second row shows the query image and its predicted result. The IoU value is reported below each image grid to facilitate performance evaluation.

extractor and optimize using the AdamW optimizer with a learning rate of $5 \times 10^{-5}$ and a batch size of 64. For our conformal prediction-based selection strategy, while acknowledging its sensitivity to the chosen quantile, we consistently set $\alpha = 0.85$, corresponding to an 85% confidence level across all tasks, and adopt the negative KL Divergence as our conformity function. We benchmark our proposed RH-Partial2Global against five previous methods: MAE-VQGAN, the unsupervised and supervised variants of VPR, the original Partial2Global framework, and prompt-SelF [11], a method primarily characterized by its ensemble-based strategy. For fair comparison, results for RH-Partial2Global are presented both with and without the test-time voting ensemble.

## 4.1 Main results

The quantitative results for the foreground segmentation, object detection and image colorization tasks are presented in Table 2. Here are several observations. First of all, RH-Partial2Global demonstrates consistent performance improvements across all visual tasks compared to the baseline, which supports the efficacy of our proposed reliable selection and holistic sampling strategies. For example, on the third fold of Pascal5$^i$, RH-Partial2Global outperforms pure Partial2Global by 0.81% and achieves an average increase of 0.62% across all four folds. Although these increments are not uniformly large, they are particularly noteworthy due to their consistent improvements across all cases without additional model training, and their confirmed statistical significance. Secondly, a nuanced observation is that the performance improvements on Fold-0 and Fold-3 are not as significant as those observed on Fold-1 and Fold-2. This disparity might be attributable to the characteristics of conformal prediction that its effectiveness in reliably predicting interval or set typically benefits from a sufficiently large calibration set. As indicated in Table 1, Fold-0 and Fold-3 contain considerably fewer samples than Fold-1 and Fold-2, potentially impacting the robustness of the reliability assessment. Lastly, the pattern of improvement remains consistent whether a test-time voting strategy is employed or not, further underscoring the generalization capability of our proposed method.

Table 3: Average top-k oracle in-context learning performances of the initial and refined alternative set on the segmentation task, which is represented by "✗" and "✓", respectively. "–" denotes the difference between the two performances.

| Top-k | Fold-0 | | | Fold-1 | | | Fold-2 | | | Fold-3 | | | Avg. | | |
|---|---|---|---|---|---|---|---|---|---|---|---|---|---|---|---|
| | ✗ | ✓ | – | ✗ | ✓ | – | ✗ | ✓ | – | ✗ | ✓ | – | ✗ | ✓ | – |
| 5 | 45.93 | 45.68 | -0.25 | 49.78 | 49.61 | -0.17 | 46.54 | 46.18 | -0.26 | 44.63 | 44.38 | -0.25 | 46.72 | 46.46 | -0.26 |
| 10 | 43.82 | 43.58 | -0.27 | 47.42 | 47.27 | -0.15 | 44.09 | 43.73 | -0.36 | 41.49 | 41.27 | -0.22 | 44.21 | 43.96 | -0.25 |
| 15 | 42.24 | 42.01 | -0.23 | 45.70 | 45.56 | -0.14 | 42.22 | 41.83 | -0.39 | 39.27 | 39.03 | -0.24 | 42.36 | 42.11 | -0.25 |

Table 4: The impact of different strategies on the segmentation task. $\mathcal{S}_{cp}$, $\mathcal{S}_{cd}$, and $\mathcal{S}_{fill}$ represent our proposed conformal prediction-guided candidate selection strategy, covering design-based sampling strategy and auxiliary filling strategy, respectively.

| | Strategy | | | Seg. (mIoU) ↑ | | | | |
|---|---|---|---|---|---|---|---|---|
| | $\mathcal{S}_{cp}$ | $\mathcal{S}_{cd}$ | $\mathcal{S}_{fill}$ | Fold-0 | Fold-1 | Fold-2 | Fold-3 | Avg. |
| (a) | ✗ | ✗ | ✗ | 38.81 | 41.54 | 37.25 | 36.01 | 38.40 |
| (b) | ✓ | ✗ | ✗ | 39.05 | 41.89 | 37.81 | 36.35 | 38.78 |
| (c) | ✗ | ✓ | ✗ | 39.15 | 41.93 | 37.75 | 36.32 | 38.79 |
| (d) | ✓ | ✓ | ✗ | **39.25** | **42.15** | **38.06** | **36.60** | **39.02** |
| (e) | ✓ | ✓ | ✓ | 39.36 | 42.54 | 38.45 | 36.72 | 39.27 |

In order to further illustrate the superiority of RH-Partial2Global, Figure 1 presents comparative visualizations of in-context examples selected by our method versus Partial2Global for the segmentation task, alongside their resulting segmentation outputs. A key observation is that, compared to Partial2Global, the prompts chosen by RH-Partial2Global generally exhibit not only high categorical relevance to the query but also greater alignment in terms of object pose, scene context, and other fine-grained visual attributes. For instance, when presented with a query image of a dog (fourth example in Fig. 1), both methods select a dog as an in-context prompt. However, RH-Partial2Global selects an image where the dog's pose precisely mirrors that of the query. Similarly, in the last visualized instance, RH-Partial2Global selects a prompt featuring a long dining table with a similar orientation to the query, while Partial2Global opts for a square table, a choice that is semantically related but structurally less analogous. These examples suggest that RH-Partial2Global is more adept at identifying in-context examples with high spatial and structural similarity to the query, which enhances their reliability and likely contributes to its superior performance across various tasks.

## 4.2 Ablation study

In order to fully validate the effectiveness of RH-Partial2Global, we conduct a series of ablation studies on it. All experiments in this section are performed on the foreground segmentation task.

**Whether RH-Partial2Global really preserves reliable examples with good quality?** A key objective is to validate whether our proposed conformal prediction-based selection strategy effectively preserves reliable, high-quality in-context examples while discarding less suitable ones. While our strategy is designed to ensure that selected examples meet a pre-defined reliability threshold, we empirically verify its impact on the qualities of the selected examples for the test set. The refined sets are obtained using our selection strategy with a confidence level $\alpha = 0.85$, which results in approximately 15% of the initial candidates being discarded as less reliable. As shown in Table 3, we evaluate the average IoU achieved by the top-5, top-10, and top-15 highest-quality examples from both the initial and the refined alternative sets. These results reveals that the performance upper bound, indicated by the average IoU of these top-k examples, remains remarkably stable, which is achieved despite discarding approximately 15% of the initial candidates. This outcome strongly suggests that our conformal prediction-based selection strategy effectively preserves high-quality, reliable examples while successfully filtering out sub-optimal ones.

**Visualization of scatter plot with regression line for similarity and IoU scores.** To intuitively illustrate the relationship between visual similarity scores and IoU scores when a specific sample serves as an in-context prompt for others, we employ scatter plots with fitted linear regression lines. Figure 2 presents several such illustrative examples. Across these visualizations, the p-values associated with the linear regression analyses are consistently below the 0.05 significance level. However, the magnitudes of the regression slopes are predominantly modest. These observations indicate that

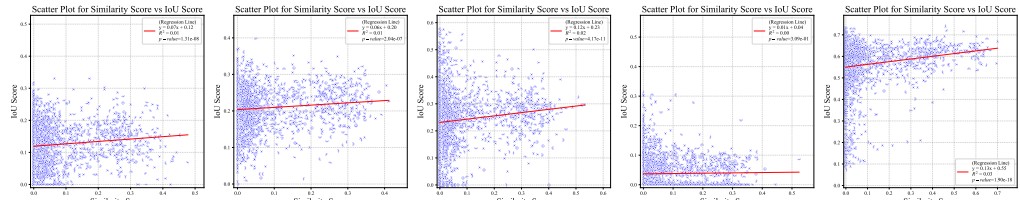

Figure 2: Visualization of scatter plot with regression line of similarity and IoU scores.

Table 5: Average top-k oracle in-context learning performances of the initial and filled alternative set on the segmentation task, which is represented by "✗" and "+", respectively. "–" denotes the difference between the two performances.

| Top-k | Fold-0 | | | Fold-1 | | | Fold-2 | | | Fold-3 | | | Avg. | | |
|---|---|---|---|---|---|---|---|---|---|---|---|---|---|---|---|
| | ✗ | + | – | ✗ | + | – | ✗ | + | – | ✗ | + | – | ✗ | + | – |
| 5 | 45.93 | 46.04 | +0.11 | 49.78 | 50.66 | +0.88 | 46.54 | 46.82 | +0.28 | 44.63 | 44.96 | +0.33 | 46.72 | 47.12 | +0.40 |
| 10 | 43.82 | 44.05 | +0.23 | 47.42 | 48.36 | +0.94 | 44.09 | 44.47 | +0.38 | 41.49 | 41.96 | +0.47 | 44.21 | 44.71 | +0.50 |
| 15 | 42.24 | 42.56 | +0.32 | 45.70 | 46.70 | +1.00 | 42.22 | 42.70 | +0.48 | 39.27 | 39.81 | +0.54 | 42.36 | 42.94 | +0.58 |

while a linear trend between visual similarity and IoU performance often exists, the strength of this linear correlation is generally weak. These visual findings corroborate the statistical conclusions from the hypothesis tests detailed in Section 3.1, further underscoring the limited predictive power of raw visual similarity for determining prompt effectiveness.

**Discussions on the effect of each strategy.** To verify the individual and combined efficacy of our proposed conformal prediction-guided selection strategy $\mathcal{S}_{cp}$ and covering design-based sampling strategy $\mathcal{S}_{cd}$, we conducted ablation experiments evaluating different configurations of these components. As shown in Table 4 (e.g., lines c and d), applying either $\mathcal{S}_{cp}$ or $\mathcal{S}_{cd}$ individually yields performance improvements over the baseline Partial2Global. Moreover, it can be seen in Table 4 that the simultaneous use of these two strategies fails to achieve a synergistic effect. A potential explanation for this observation is that $\mathcal{S}_{cp}$ can occasionally reduce the number of reliable candidates to fewer than the ranker length $k$. Under such circumstances, $\mathcal{S}_{cd}$ loses its intended utility, as any sampling can ensure complete coverage of all pairwise preferences. Therefore, we introduce an auxiliary filling strategy $\mathcal{S}_{fill}$, selecting the most similar candidates from the constructed reliable set $\mathcal{Y}_{\alpha}$ to the query sample $x_q$ for filling. As presented in the line (e) of Table 4, this strategy can further enhance the performance of our RH-Partial2Global. We attribute this enhancement to an improved performance upper bound. Similarly, we directly test all prompts from the original and the filled alternative set for each query in the segmentation task and present the average performance of the top-5/10/15 best examples in Table 5. The improvements in this oracle performance, as shown in Table 5, lend strong support to our viewpoint.

**Universality of conformal prediction-guided selection strategy.** The similarity-priority assumption, which our proposed conformal prediction-guided selection strategy $\mathcal{S}_{cp}$ aims to address, represents a common limitation in many contemporary Visual In-Context Learning (VICL) methods, extending beyond just the Partial2Global framework. To demonstrate the universality and generalization capacity of $\mathcal{S}_{cp}$, we therefore apply it to both the unsupervised (UnsupPR) and supervised (SupPR) variants of the VPR framework [2]. Recognizing that $\mathcal{S}_{cp}$ applied alone might occasionally reduce the number of reliable candidates to even zero, we implemented and evaluated two augmented approaches to ensure a viable candidate pool when applying our strategy to VPR: (1) $\mathcal{S}_{cp}$ *with initial supplementation (referred to as "+$\mathcal{S}_{cp}$").* If the refined alternative set is empty, it is supplemented by selecting the most visually similar candidate from the initial alternative set provided by VPR; (2) $\mathcal{S}_{cp}$ *combined with the auxiliary filling strategy $\mathcal{S}_{fill}$ (referred to as "+$\mathcal{S}_{cp}$+$\mathcal{S}_{fill}$").* If the resulting alternative set is empty, it is augmented by selecting the most visually similar candidate from the reliable global set $\mathcal{Y}_{\alpha}$. We utilize similarity scores obtained from VPR's pretrained and fine-tuned metric network respectively in Eq.(2). The confidence level $\alpha$ is set as 0.55 across all the four folds in the segmentation task. We present all the comparison results in Table 6.

Here are several observations. Firstly, our proposed conformal prediction-guided selection strategy can also demonstrate significant and consistent performance gains across all the folds when compared to the baseline VPR. This outcome strongly validates the effectiveness and broader universality of $\mathcal{S}_{cp}$. Moreover, our auxiliary filling strategy $\mathcal{S}_{fill}$ is shown to further enhance the performance of VICL, underscoring the quality of our reliable prompt set identified by $\mathcal{S}_{cp}$.

Table 6: Performance comparison of VPR variants (UnsupPR and SupPR) with and without the integration of $\mathcal{S}_{cp}$ and its augmentation $\mathcal{S}_{fill}$ on the segmentation task.

| Method | Ref. | Seg. (mIoU) ↑ | | | | |
| --- | --- | --- | --- | --- | --- | --- |
| | | Fold-0 | Fold-1 | Fold-2 | Fold-3 | Avg. |
| UnsupPR [2] | NIPS'23 | 34.75 | 35.92 | 32.41 | 31.16 | 33.56 |
| UnsupPR+$\mathcal{S}_{cp}$ | – | 36.07 | 38.50 | **35.03** | 33.53 | 35.78 |
| UnsupPR+$\mathcal{S}_{cp}$+$\mathcal{S}_{fill}$ | – | **36.97** | **38.80** | 34.62 | **33.66** | **36.01** |
| SupPR [2] | NIPS'23 | 37.08 | 38.43 | 34.40 | 32.32 | 35.56 |
| SupPR+$\mathcal{S}_{cp}$ | – | 37.69 | 39.45 | 36.61 | 33.96 | 36.93 |
| SupPR+$\mathcal{S}_{cp}$+$\mathcal{S}_{fill}$ | – | **37.83** | **39.85** | **36.61** | **33.96** | **37.06** |

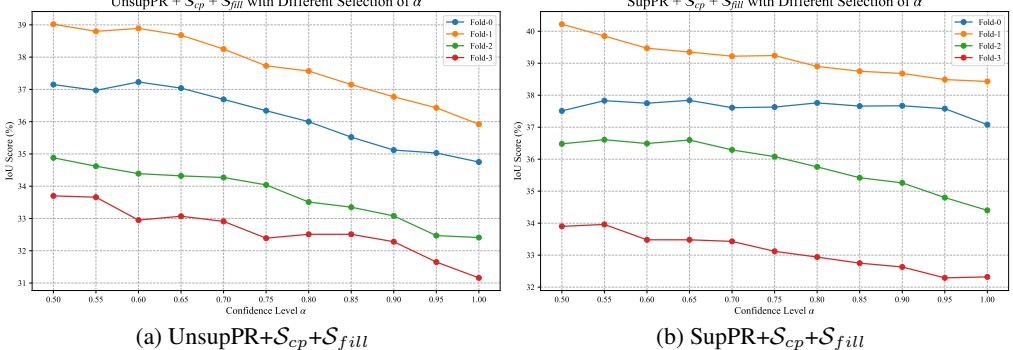

(a) UnsupPR+$\mathcal{S}_{cp}$+$\mathcal{S}_{fill}$      (b) SupPR+$\mathcal{S}_{cp}$+$\mathcal{S}_{fill}$

Figure 3: Performance trends of VPR when enhanced by the $\mathcal{S}_{cp}$ and $\mathcal{S}_{fill}$ strategies, evaluated across a range of $\alpha$ parameter values. The setting '$\alpha = 1.00$' represents the baseline performance of the original VPR methods without these proposed strategies.

Figure 3 further details the performance of SupPR and UnsupPR when integrated with our $\mathcal{S}_{cp}$ and $\mathcal{S}_{fill}$ strategies, evaluated under various settings of the conformal prediction parameter $\alpha$. It is evident from these results that regardless of the specific $\alpha$ value selected, applying our strategies enables both VPR variants to consistently outperform the original VPR framework. Such consistent outperformance, even with varying $\alpha$ configurations, strongly demonstrates the universality and robustness of our conformal prediction-guided selection strategy.

## 5 Conclusion

This paper introduces RH-Partial2Global, an enhanced variant of Partial2Global, tailored for in-context example selection in Visual In-Context Learning (VICL). Specifically, we first challenge the default similarity-priority assumption that an image more similar to the query image is more suitable as an in-context example, and further validate its inherent limitations from a statistical perspective. Subsequently, we develop a sample selection strategy based on jackknife conformal prediction to refine the alternative set, which is initially established following the similarity-priority assumption, thereby retaining only reliable candidate samples. Furthermore, we propose a covering design-based sampling strategy to supersede the random operations in Partial2Global, facilitating a more comprehensive and balanced construction of pairwise preference relationships. By integrating these two strategies, our RH-Partial2Global yields improved global ranking predictions, paving the way for more reliable and holistic VICL prompt selection. Extensive experiments across multiple visual tasks demonstrate that RH-Partial2Global not only outperforms its predecessor, Partial2Global, but also consistently achieves excellent performance.

**Limitations.** While our proposed method significantly enhances Partial2Global, its reliable set prediction is sensitive to dataset size, potentially limiting gains with scarce data due to reduced statistical robustness. Nevertheless, we contend that our reflections on the similarity-priority assumption and exploration based on conformal prediction hold broader significance for VICL. This insight can encourage the development of example selection strategies that move beyond mere similarity, potentially advancing performance across diverse models and tasks.

**Acknowledgments**

This research is partly supported by the Hubei Provincial Natural Science Foundation of China under Grant No.2022CFA055.

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

# A Addition ablation study and discussion

**Analysis of confidence level $\alpha$ for RH-Partial2Global.** To further demonstrate the robustness of our proposed strategy with respect to the confidence level $\alpha$, we conduct a corresponding ablation study on RH-Partial2Global, evaluating its performance with $\alpha$ values ranging from 0.94 to 0.80. The results are presented in Figure 4. The results demonstrate that all tested $\alpha$ values yield a performance improvement over the baseline (i.e. $\alpha = 1.00$).

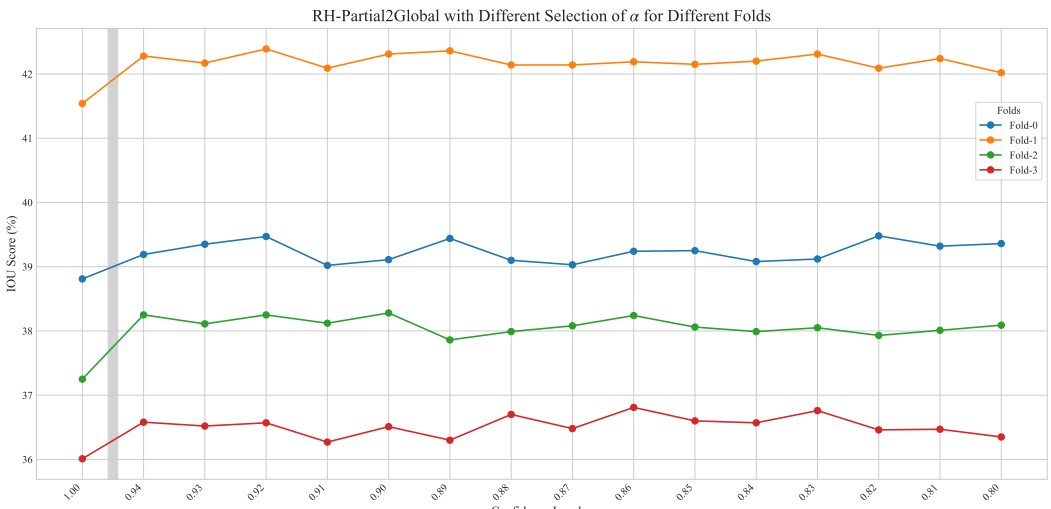

Figure 4: Performance trends of PH-Partial2Global, evaluated across a range of $\alpha$ parameter values. The setting '$\alpha = 1.00$' represents the baseline performance of the pure Partial2Global method.

**The selection of our conformity function.** To identify the more effective conformity function for our strategy, we evaluate both Spearman correlation and negative KL Divergence beyond upon the Partial2Global framework, comparing their respective impacts on overall performance. The comparison results are shown in Table 7.

Table 7: Comparison of our proposed RH-Partial2Global with Spearman correlation or negative KL Divergence as the conformity function.

| Method | Ref. | Seg. (mIoU) ↑ | | | | |
|---|---|---|---|---|---|---|
| | | Fold-0 | Fold-1 | Fold-2 | Fold-3 | Avg. |
| Partial2Global [6] | NIPS'24 | 38.81 | 41.54 | 37.25 | 36.01 | 38.40 |
| RH-Partial2Global (Cor) | – | 39.13 | 41.88 | 37.57 | 36.48 | 38.77 |
| RH-Partial2Global (KL) | – | **39.25** | **42.15** | **38.06** | **36.60** | **39.02** |

Experimental results on the Pascal-$5^i$ dataset indicate that negative KL Divergence consistently yields superior segmentation performance compared to Spearman correlation across all the four folds. Based on this finding, negative KL Divergence is adopted as the definitive conformity function for our main experiments. This observed advantage can be attributed to the fundamental differences in how these metrics capture data characteristics. Spearman correlation, which quantifies rank-based monotonic associations, primarily reflects ordinal structure. Consequently, it is less sensitive to the detailed distributional information embedded in specific value magnitudes and higher-order statistical moments of the underlying data. In contrast, KL Divergence assesses the dissimilarity between entire probability distributions, thereby inherently accounting for discrepancies across all orders of moments and offering a more comprehensive comparison of distributional differences.

**Visualization of single object detection task.** Figure 5 presents comparative visualizations of in-context examples selected by our method versus Partial2Global for the detection task, alongside their resulting detection outputs. Consistent with findings from the segmentation task, prompts selected by RH-Partial2Global generally exhibit superior alignment with the query in terms of scene context and other fine-grained visual attributes when compared to those from Partial2Global. For instance, as illustrated in the central column of comparative visualizations, the scene context within the prompts chosen by our RH-Partial2Global demonstrates a notably higher degree of similarity to the query sample's context.

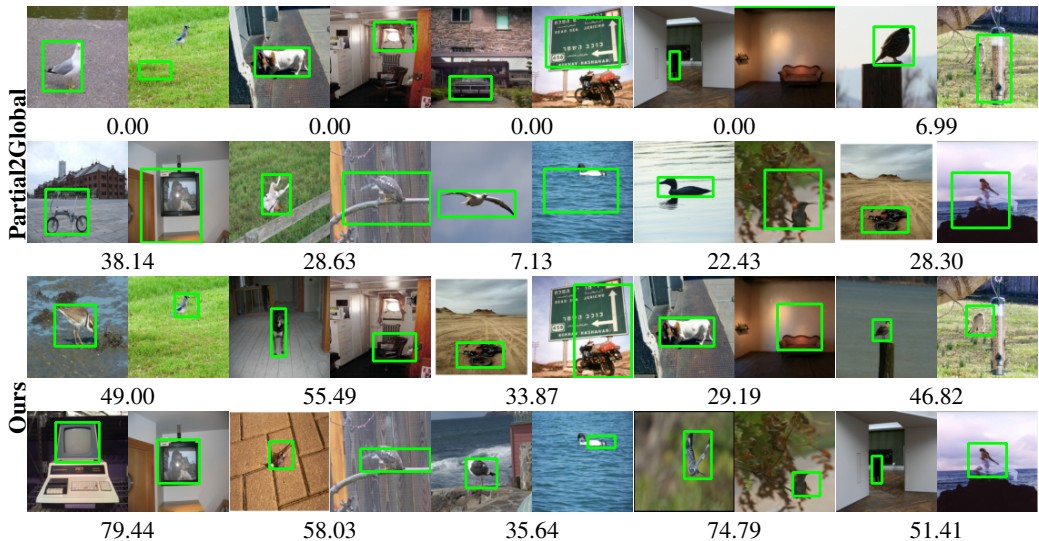

Figure 5: Qualitative comparison of single object detection performance between our RH-Partial2Global and Partial2Global. To enhance visual clarity and simplicity, bounding boxes are overlaid directly onto the images, rather than displaying the complete image grids. Each depicted item consists of an in-context example (left) and the corresponding query image (right).

**Why the selection strategy is jackknife conformal prediction-guided?** The underlying reasons can be summarized as follows. Firstly, mirroring the leave-one-out strategy of jackknife, our selection strategy individually evaluates each candidate by assessing its performance when used as a prompt for the other samples. Secondly, similar to jackknife conformal prediction, our selection strategy emphasized maintaining reliability across the alternative set rather than optimizing for a single sample. Lastly, instead of pinpointing a single best prompt, it selects a set of reliable prompts, aligning with conformal prediction's principle of "set prediction". Therefore, our selection strategy can be regarded as a jackknife conformal prediction-guided method.

# B   Broader impact

The proposed work itself is not anticipated to lead to significant direct negative social impacts. Despite of this, it explicitly acknowledges and discusses a critical broader concern: data bias. The paper notes that if the data used for in-context learning (or any data-driven AI model) contains existing societal biases, the models leveraging this data (including those incorporating the proposed enhancements) could inadvertently perpetuate or even amplify these biases. This could potentially lead to unfair or discriminatory outcomes, especially in sensitive application areas.

