# OpenReview forum: "Towards Reliable and Holistic Visual In-Context Learning Prompt Selection"
_NeurIPS.cc/2025/Conference — NeurIPS 2025 poster_

### Official Review · Reviewer_jvC8 · 2025-07-02

**Clarity:** 2
**Significance:** 2
**Originality:** 2
**Rating:** 3
**Confidence:** 3

**Summary:**

The paper proposes RH-Partial2Global, a drop-in replacement for the widely used Partial2Global prompt-selection pipeline in Visual In-Context Learning (VICL). The authors first show that the common “pick the K most visually similar images” heuristic only weakly predicts prompt quality, and that the usual random shuffling of candidates leads to redundant, incomplete pairwise comparisons. To address these issues they introduce two training-free modules: (i) a reliable selection stage that uses jackknife conformal prediction to keep only those candidate prompts whose quality is statistically consistent with their similarity scores, and (ii) a holistic sampling stage that replaces random shuffles with a combinatorial covering design guaranteeing every image pair is compared at least once while minimising duplicates.

**Questions:**

1. Given the marginal improvement, can you report mean ± sd over several random seeds to substantiate the observed gains?
2. Have you tried RH-Partial2Global on classification-only VICL or on video prompt selection?
3. Would it benefit multi-object detection?

**Ethical Concerns:**

["NO or VERY MINOR ethics concerns only"]

**Final Justification:**

Given the marginal improvement and dataset size sensitivity, I would keep my rating.

**Limitations:**

Except for the weaknesses mentioned above, this work's generality is still narrow; all tasks are image-level, no tests on video, multi-object detection, or classification-only settings.

**Paper Formatting Concerns:**

No major issues.

**Quality:**

2

**Strengths And Weaknesses:**

Strengths:
1. Introduces conformal prediction and covering-design theory to VICL in a principled way.
2. This method can be integrated into any Partial2Global-style method without requiring retraining of the backbone or ranker.
3. The paper is well-written and provides algorithm pseudocode.

Weaknesses:
1. The biggest concern is that this work has modest absolute gains – improvements are <1%.
2. This method is sensitive on dataset size, folds with fewer samples benefit less.

---

> ### Author Rebuttal · Authors · 2025-07-30
>
> ### **For Weakness 1: Regarding the performance gains**
>
> We thank the reviewer for this excellent and critical question. We agree that a deeper analysis of the performance gains, particularly for the cases where they appear modest on the Partial2Global baseline, is both necessary and beneficial. Our analysis suggests that the magnitude of the performance gain is influenced by several factors.
>
> 1. **We would like to highlight that our method's impact varies depending on the baseline it is applied to.** As detailed in the **Table 1 of Supplementary Material**, our strategy yields substantial improvements when integrated with other baselines, such as the VPR variants, where we observed gains ranging from **1.5% to 3%**. This demonstrates the general efficacy and high potential of our proposed method under different conditions.
>
> 2. **Moreover, in our initial main experiments, we deliberately used a single, fixed confidence level ($\alpha=0.85$) across all tasks.** This was to demonstrate the "out-of-the-box" utility and ease of use of our method without task-specific tuning. To test the hypothesis that gains could be larger with a tuned $\alpha$, we conducted a new experiment as follows.
>
>    |    Fold-0     |    Fold-1     |    Fold-2     |    Fold-3     | Avg.  |
>    | :-----------: | :-----------: | :-----------: | :-----------: | :---: |
>    | $\alpha=0.85$ | $\alpha=0.85$ | $\alpha=0.85$ | $\alpha=0.85$ |  --   |
>    |     39.25     |     42.15     |     38.06     |     36.60     | 39.02 |
>    | $\alpha=0.82$ | $\alpha=0.92$ | $\alpha=0.90$ | $\alpha=0.86$ |  --   |
>    |     39.48     |     42.39     |     38.28     |     36.81     | 39.24 |
>
>    The results show that when $\alpha$ is selected on a per-dataset basis, the performance gains on Partial2Global become more significant.
>
> 3. **Furthermore, MAE-VQGAN, the common fundamental inpainting model used in current VICL methods like VPR and Partial2Global, is a little out-of-date.** Therefore, although RH-Partial2Global is highly optimized, it operates near the ceiling for MAE-VQGAN.
>
>    **To validate our opinion, we standardize the backbone architecture by replacing MAE−VQGAN with SegGPT [1], a more advanced VICL model in both the Partial2Global baseline and our proposed RH-Partial2Global framework.** Since the prompt selection mechanism in SegGPT involves randomly sampling images that share the same category label from the training set of PASCAL-5i as the prompt image (as illustrated in its Appendix A), we contend that this approach constitutes a form of information leakage, as it implicitly provides the model with class-level knowledge. Consequently, to create a stricter and more fair baseline, we utilize the single most similar image as the prompt and evaluate under our settings. This allows the evaluation to focus solely on the merits of our proposed strategies. The quantitative results for segmentation task are presented as follows.
>
>    |           Method           | Fold-0 | Fold-1 | Fold-2 | Fold-3 | Avg.  |
>    | :------------------------: | :----: | :----: | :----: | :----: | :---: |
>    |           SegGPT           | 64.35  | 66.79  | 58.91  | 65.68  | 63.93 |
>    |  Partial2Global (SegGPT)   | 66.52  | 71.55  | 62.48  | 68.74  | 67.32 |
>    | RH-Partial2Global (SegGPT) | 69.45  | 74.35  | 66.33  | 71.90  | 70.51 |
>
>    Notably, by integrating a more advanced generative model, our RH-Partial2Global achieves an average performance improvement of over **3%** over Partial2Global.
>
>    [1] SegGPT: Segmenting Everything In Context
>
> ------
>
> ### **For Weakness 2: Regarding the sensitivity to dataset size**
>
> We agree with this observation and, in the spirit of transparency, had already identified and discussed this sensitivity to dataset size in the **Limitations** section of our original manuscript.
>
> **Regarding this phenomenon, we provide a principled analysis grounded in the theory of conformal prediction: The effectiveness of conformal prediction relies on the exchangeability of data, which is more robust with larger datasets.** With smaller sets (as in our Fold-0 and Fold-3), two issues may arise: (i) The empirical distribution of conformity scores becomes more coarse-grained, making the quantile estimation less precise. (ii) The set of scores is more susceptible to the influence of outliers or sampling noise, which can destabilize the calibration process. Recognizing the importance of this issue, we will conduct a more in-depth investigation into the interplay between dataset size, hyperparameter choice, and performance in our future research.
>
> **Nevertheless, we wish to emphasize that despite this inherent sensitivity in low-data regimes, our proposed selection strategy remains a powerful and generally applicable contribution.** The primary goal of our work is to introduce a more principled approach to candidate selection, and its broad effectiveness is demonstrated by the performance gains of both on VPR, Partial2Global and SegGPT. This indicates that while the magnitude of the benefit is moderated by dataset size, a known characteristic of conformal methods, the overall value and utility of our framework across different model architectures are robust.
>
> ------
>
> ### **For Question 2: Regarding the presentation of standard deviation**
>
> The primary source of randomness in our experiments stems from the sampling step within the aggregation process of the Partial2Global baseline. To ensure the stability of our results, the performance metrics reported in our original manuscript were, in fact, already the **average of 10 independent runs**.
>
> We have now calculated the standard deviation from these 10 runs. The results are now presented in the **mean ± standard deviation** format as follows.
>
> |      method       |     Fold-0      |     Fold-1      |     Fold-2      |     Fold-3      |
> | :---------------: | :-------------: | :-------------: | :-------------: | :-------------: |
> | RH-Partial2Global | 39.25$\pm$0.097 | 42.15$\pm$0.098 | 38.06$\pm$0.100 | 39.25$\pm$0.098 |
>
> This tables show that the standard deviations are consistently small, confirming that the observed gains, while sometimes modest, are statistically stable and reproducible, rather than being an artifact of random variation. We believe this addition significantly strengthens the empirical evidence for our method's effectiveness.
>
> ------
>
> ### **For Question 2, 3 and Limitations: Regarding the generality of our work**
>
> We thank the reviewer for these insightful suggestions regarding the application of our method to other important domains such as video, multi-object detection, and classification-only tasks. These are indeed exciting future directions for VICL research, and we appreciate the opportunity to clarify the scope of our current work and the generalizability of our proposed framework.
>
> The primary reason for focusing on image-level tasks in this study is the set of established conventions and technical limitations of the prevailing VICL frameworks.
>
> 1. **Backbone Model Constraints:** The standard foundational model used in this line of research, MAE-VQGAN, is architecturally designed for image synthesis and inpainting. which does not inherently support the processing of temporal data.
> 2. **Lack of Established Task Formulations:** For tasks like classification-only VICL, there is currently no consensus or established framework in the literature for constructing the necessary "example pairs" (akin to the image-segmentation mask pairs used in our work). Defining what constitutes an effective visual prompt for a classification label is an open research problem in itself.
>
> **Despite these current constraints, we would like to emphasize that one of the core contributions of our paper, the CP-based prompt selection strategy, is conceptually modality-agnostic.** Our framework addresses the fundamental problems of how to best construct a reliable candidate set and how to select from it, a process that is independent of whether the input data is an image, a video frame, or another data type. Therefore, we strongly believe that our framework is forward-compatible. As more powerful and versatile foundational models emerge that are capable of video-based or multi-task in-context learning, our selection strategy can be seamlessly integrated to enhance their prompting efficiency and reliability. The principles of leveraging conformal prediction to manage uncertainty and build a reliable set remain directly applicable.

---

> ### Author Response · Authors · 2025-08-06
> **Follow-up on Review for Submission #2552**
>
> Dear Reviewer jvC8,
>
> We hope this message finds you well. Thank you again for your valuable time and insightful comments on our work.
>
> We have posted our response, in which we have addressed all your valuable points. Specifically, we have provided deeper analyses of performance gains and dataset size sensitivity, updated results to a mean ± standard deviation format for statistical rigor, and expanded the discussion on the generality of our work.
>
> We wanted to gently follow up to see if our clarifications were sufficient, or if there is any additional information we can provide from our end.
>
> We remain actively engaged in the Author-Reviewer discussion period and are standing by to answer any further questions you may have.
>
> Best regards,
>
> The Authors of Submission #2552

---

### Official Review · Reviewer_rrVW · 2025-07-02

**Clarity:** 3
**Significance:** 3
**Originality:** 3
**Rating:** 4
**Confidence:** 4

**Summary:**

This paper introduces a variant framework, called  RH-Partial2Global, based on a previous work Partial2Global for in-context example selection in Visual In-Context Learning (VICL).
The authors first challenge the default similarity-priority assumption in the Partial2Global framework that similarity between images can create better in-context examples. Then they point out that the random sampling strategy of candidate examples leads to incomplete coverage of pairwise relationships. To address the two key limitations in Partial2Global framework, the authors try to build a reliable and holistic in-context example selection from a statistical perspective. Specifically, they propose two strategies-a jackknife conformal prediction-based example selection strategy and a covering design-based sampling strategy in their work. Extensive experiments show that their proposed work performs better than the baseline Partial2Global framework across diverse vision tasks.

**Questions:**

1.	For Conformal Prediction, pre-defined confidence is necessary, but different settings will affect the final performance. The paper  does not seem to include an analysis of the effect of different confidence scores in CP. Or did I miss this part?
2.	247-248 lines, ‘... incurs negligible additional computational overhead ...’, the authors should provide the exact cost burden to illustrate?
3.	In the paper, the baseline model Partial2Global is regarded as the state-of-the-art method. However there are other papers in the VICL area like: ‘Embracing Collaboration Over Competition: Condensing Multiple Prompts for Visual In-Context Learning’, it has better performance than RH-Partial2Global method. Could you explain why you do not compare with the methods?

**Ethical Concerns:**

["NO or VERY MINOR ethics concerns only"]

**Final Justification:**

The authors have addressed my concerns quite well and I am happy to increase the rating of their paper.

**Limitations:**

Yes

**Paper Formatting Concerns:**

The Figures in the paper could be better. For example, the ratios between width and length of images in Figure1 are incongruent. Also, the plots in Figure 2 are not clear enough.

**Quality:**

2

**Strengths And Weaknesses:**

Strengths:
1.	The authors offer a new perspective on reliable prompt selection for VICL. In the paper, the work is the first work to introduce Conformal Prediction (CP) to the VICL, as far as the reviewer is aware. Based on CP, a jackknife selection strategy is proposed to construct a more trustworthy alternative set of in-context examples.
2.	The authors incorporate a covering design-based sampling strategy within the consistency-awaking aggregator of Partial2Global, to have more holistic coverage of pairwise preferences and consequently with a more accurate global ranking.
3.	Extensive experiments on various datasets and tasks illustrate the performance and validity of the proposed methodology. To validate the validity of different strategies, comprehensive ablation studies were conducted, demonstrating the effectiveness of these strategies.

Weaknesses:
1.	Although the paper proposed two strategies to improve the performance in VICL, the novelty is not enough.  The authors only use the jackknife CP to refine the construction of alternative sets and replace the original random sampling in Partial2Global with a covering design-based strategy.
2.	For Conformal Prediction, pre-defined confidence is necessary, as different settings will affect the final performance. The paper does not include any analysis of the effect of different confidence scores in CP.

---

> ### Author Rebuttal · Authors · 2025-07-30
>
> ### **For Weakness 1: Regarding the novelty of our work**
>
> We thank the reviewer for the feedback on the novelty of our work. We would like to respectfully clarify that the core novelty of our work lies not only in the individual strategies themselves, but also in our strong motivation from statistical perspective.
>
> **Our first contribution is to present the first rigorous and formal challenge to the ubiquitous similarity-priority assumption that underpins most current VICL methods.** We do this not by intuition, but by conducting a statistical hypothesis test that provides quantitative evidence of this assumption's shortcomings. To our knowledge, this is the first work to so rigorously diagnose this foundational issue. This strong, data-driven motivation is the bedrock of our paper. Our main experiments also demonstrate the effectiveness of the proposed jackknife conformal prediction-guided selection strategy targeted at this motivation.
>
> **In addition, our CP-based selection strategy can be identified as a universal and highly generalizable method.** please note that we also conducted extensive experiments on the VPR baseline to validate the universality of CP-guided selection strategy, achieving performance gains of **1.5%** to **3%** in our **Supplementary Material** **(Table 1)**. These results demonstrate that our CP-based selection strategy is not a narrow trick for one specific method but a **generalizable contribution** that can advance how the entire VICL field approaches candidate set construction.
>
> ------
>
> ### **For Weakness 2 and Question 1: Regarding the analysis of confidence level**
>
> We sincerely thank the reviewer for this critical and highly constructive point. We agree that a deeper exploration of our method's sensitivity to the calibration set size, and its interaction with the confidence level $\alpha$, is essential for understanding its practical applicability and limitations.
>
> **To demonstrate the robustness of our proposed strategy with respect to the confidence level $\alpha$, we present the following ablation studies on VPR and Partial2Global.** We would like to draw your attention to the ablation study in the **Supplementary Material (Figure 1)**. This study, conducted on the VPR baseline, shows that our strategy consistently delivers performance improvements across a wide range of $\alpha$ values (from 0.5 to 0.95). This result establishes the general robustness and effectiveness of our core approach. *(Note: We used VPR for this specific analysis because its architecture provides a cleaner and more direct assessment of $\alpha$'s impact, allowing for $\alpha$ more isolated analysis of the hyperparameter itself.)*
>
> **For comprehensive analysis, we conduct a corresponding ablation study on RH-Partial2Global**, evaluating its performance with $\alpha$ values ranging from 0.96 to 0.80. The results are presented below.
>
> | $\alpha$ | 1.00  | 0.94  | 0.93  |   0.92    | 0.91  |   0.90    | 0.89  | 0.88  | 0.87  |   0.86    | 0.85  | 0.84  | 0.83  |   0.82    | 0.81  | 0.80  |
> | :------: | :---: | :---: | :---: | :-------: | :---: | :-------: | :---: | :---: | :---: | :-------: | :---: | :---: | :---: | :-------: | :---: | :---: |
> |  Fold-0  | 38.81 | 39.19 | 39.35 |   39.47   | 39.02 |   39.11   | 39.44 | 39.10 | 39.03 |   39.24   | 39.25 | 39.08 | 39.12 | **39.48** | 39.32 | 39.36 |
> |  Fold-1  | 41.54 | 42.28 | 42.17 | **42.39** | 42.09 |   42.31   | 42.36 | 42.14 | 42.14 |   42.19   | 42.15 | 42.20 | 42.31 |   42.09   | 42.24 | 42.02 |
> |  Fold-2  | 37.25 | 38.25 | 38.11 |   38.25   | 38.12 | **38.28** | 37.86 | 37.99 | 38.08 |   38.24   | 38.06 | 37.99 | 38.05 |   37.93   | 38.01 | 38.09 |
> |  Fold-3  | 36.01 | 36.58 | 36.52 |   36.57   | 36.27 |   36.51   | 36.30 | 36.70 | 36.48 | **36.81** | 36.60 | 36.57 | 36.76 |   36.46   | 36.47 | 36.35 |
>
> The results demonstrate that all tested $\alpha$ values yield a performance improvement over the baseline (i.e. $\alpha=1.00$).
>
> ------
>
> ### **For Question 2: Regarding the exact cost burden**
>
> **The additional computational cost of our method stems from a one-time, offline preprocessing step: the construction of the conformity score sequence.** This process, which involves running inference on the training set, is highly parallelizable and introduces a modest overhead, equivalent to approximately 5% of the baseline method, Partial2Global's total training time when measured on the same hardware.
>
> ------
>
> ### **For Question 3: Regarding the comparison with *Condensation***
>
> We thank the reviewer for bringing the paper 'Embracing Collaboration Over Competition: Condensing Multiple Prompts for Visual In-Context Learning' (*Condensation*) to our attention. This provides a valuable opportunity to clarify the fundamental distinction between two paradigms in VICL: **retrieval-based methods**, which operate under black-box assumptions, and **finetuning-based methods** that require white-box access.
>
> **On one hand, our  work, alongside VPR and Partial2Global, belongs to the retrieval-based paradigm and is designed to operate in a strict black-box setting.** These methods treat the core generative model as an opaque system, relying only on weak supervision signals such as final ranking scores. They require no knowledge of the model's internal architecture, weights, or intermediate features. This black-box formulation is notably more challenging but also more practical and general, especially as VICL increasingly involves large, proprietary foundation models accessible only via APIs.
>
> **On the other hand, the finetuning-based method like InMeMo [1] and the mentioned *Condensation* necessitates white-box (or at least grey-box) access.** They necessitate direct access to and manipulation of intermediate representations or even require backpropagation of gradients through the model, which constitutes a form of privileged information that is unavailable in a true black-box scenario. **Given that our method operates under a more constrained and challenging black-box assumption while *Condensation* leverages the advantages of a white-box setting, a direct performance comparison would be inequitable.**
>
> **Furthermore, the two paradigms are not mutually exclusive but potentially synergistic.** We note that *Condensation* also begins by constructing an initial candidate through similarity priority assumption (as illustrated in **its Section 3.1**). Our principled, CP-based selection strategy could therefore be employed as a more robust front-end to enhance this method, providing a higher-quality set of candidates for its condensation process.
>
> [1] Instruct Me More! Random Prompting for Visual In-Context Learning
>
> -----
>
> ### **For Paper Formatting Concerns:**
>
> We thank the reviewer for their valuable feedback regarding the formatting and clarity of our figures. For Figure 1, we appreciate the reviewer pointing out the inconsistent aspect ratios. This was an artifact of the model's preprocessing pipeline, as the MAE-VQGAN backbone requires all input images to be resized to a fixed 112x112 resolution for concatenation, which we depicted literally in the original figure. For Figure 2, we will increase the font size for all axes, labels, and legends to improve readability.

---

> > ### Comment · Reviewer_rrVW · 2025-08-07
> > **The additional experimental results demonstrate the robustness of their method to the setting of confidence level. The arguments regarding the applicability of their approach to black box settings is convincing.**
> >
> > Thank you for the detailed rebuttal. The authors provide good arguments pointing out the key differences in the assumption underpinning their approach as compared with condensation. The results demonstrating robustness of their method to hyper parameter alpha are appreciated.

---

> > > ### Author Response · Authors · 2025-08-07
> > > **Thank you for your feedback.**
> > >
> > > Dear Reviewer rrVW,
> > >
> > > Thank you very much for re-evaluating our work and for your encouraging feedback. We are pleased to know that our clarifications have addressed your questions.
> > >
> > > We truly appreciate the time and expertise you have dedicated to our paper. Thanks in large part to your constructive suggestions, we feel the manuscript is substantially stronger and clearer now.
> > >
> > > We hope the improved version of the manuscript will be considered a strong contribution to the community.
> > >
> > > Best regards,
> > >
> > > The Authors of Submission #2552

---

> ### Author Response · Authors · 2025-08-06
> **Follow-up on Review for Submission #2552**
>
> Dear Reviewer rrVW,
>
> We hope this message finds you well. Thank you again for your valuable time and insightful comments on our work.
>
> We have posted our response, in which we have addressed all your valuable points. Specifically, we have clarified the core novelty of our framework, provided deeper analyses of the confidence level and computational cost, discussed the comparison with methods like *Condensation*, and revised all figures for enhanced clarity.
>
> We wanted to gently follow up to see if our clarifications were sufficient, or if there is any additional information we can provide from our end.
>
> We remain actively engaged in the Author-Reviewer discussion period and are standing by to answer any further questions you may have.
>
> Best regards,
>
> The Authors of Submission #2552

---

### Official Review · Reviewer_LDet · 2025-07-03

**Clarity:** 2
**Significance:** 2
**Originality:** 3
**Rating:** 4
**Confidence:** 2

**Summary:**

This paper introduces an enhanced variant of Partial2Global designed for the reliable and holistic selection of in-context examples in Visual In-Context Learning. Specifically, this paper leverages a jackknife conformal prediction-guided strategy to construct reliable 16 alternative sets and a covering design-based sampling approach to ensure comprehensive and uniform coverage of pairwise preferences.

**Questions:**

I have one question (no need to address it in the response). While Visual In-Context Learning is an intriguing task, many general-purpose models (e.g., SegGPT) already demonstrate strong capabilities in this domain due to their training on large-scale data. Given this context, what is the necessity of studying Visual In-Context Learning on small datasets like Pascal-5i or Pascal VOC 2012?

Additionally, for larger models, such as those described in "Sequential Modeling Enables Scalable Learning for Large Vision Models", does investigating prompt selection strategies become more critical compared to small-data experiments? The authors should clarify whether their work can be used for existing large-scale models.

**Ethical Concerns:**

["NO or VERY MINOR ethics concerns only"]

**Final Justification:**

An enhanced variant of Partial2Global is designed for the reliable and holistic selection of in-context examples in VICLl. The experimental results across diverse visual tasks show excellent performance. My concerns are well solved in the rebuttal. After considering other reviews and the corresponding answers, I'd like to keep the rating at the current stage.

**Limitations:**

yes

**Paper Formatting Concerns:**

No issues

**Quality:**

3

**Strengths And Weaknesses:**

Strengths:
The paper is well-motivated and well-written.

An enhanced variant of Partial2Global is designed for the reliable and holistic selection of in-context examples in VICLl.

The experimental results across diverse visual tasks show excellent performance.

Weaknesses:
1. More general tasks: SegGPT[1] is a generalist model for segmenting everything in Visual In-Context
Learning. Can RH-Partial2Global be used in this task?

2. In Table 2, can Painter and SegGPT be used in this dataset?

[1] SegGPT: Segmenting Everything In Context
[2] Sequential Modeling Enables Scalable Learning for Large Vision Models

---

> ### Author Rebuttal · Authors · 2025-07-30
>
> ### **For Question 1: Regarding VICL on small datasets**
>
> We sincerely thank the reviewer for these insightful, high-level questions. We completely agree that large-scale, general-purpose models like SegGPT have demonstrated truly impressive capabilities in VICL. We believe it is a valuable opportunity to clarify the specific research gap our work aims to fill and the broader context in which it sits.
>
> **Rather than focusing on training a massive, end-to-end model, our work, along with related methods like VPR and Partial2Global, formulates VICL as a prompt selection problem, which is also a critical issue for general-purpose models.** From this perspective, the role of smaller, standardized datasets like Pascal-5i and VOC 2012 becomes critical, but in a different capacity. These datasets are not used to train the foundational models (our base inpainting model, MAE-VQGAN, was pre-trained on a larger dataset). Instead, they serve as controlled, reproducible **benchmarks**. Their purpose is to enable a rigorous and fair evaluation of the prompt selection algorithms themselves, allowing the research community to isolate and compare the efficacy of different selection strategies.
>
> ----
> ### **For Weaknesses and Question 2: Regarding the applicability of our work to large-scale models like SegGPT**
>
> We thank the reviewer for this insightful question regarding the applicability of our work to large-scale models. This is a critical point that touches upon the scalability and future relevance of our framework
>
> **In fact, our method treats the generative or in-context learning model as a black box, where the generative model in our framework, MAE-VQGAN is a modular component for fair comparison, which could be readily swapped for any state-of-the-art inpainting model.** The fundamental question we address, "how does one best select visual prompts to elicit a desired behavior from a pre-trained model through its in-context learning ability?", remains crucial and unsolved, regardless of whether the base model is MAE-VQGAN or a future large-scale foundation model.
>
> **To validate our opinion, we standardize the backbone architecture by replacing MAE−VQGAN with SegGPT in both the Partial2Global baseline and our proposed RH-Partial2Global framework.** Since the prompt selection mechanism in SegGPT involves randomly sampling images that share the same category label from the training set of PASCAL-5i as the prompt image (as illustrated in its Appendix A), we contend that this approach constitutes a form of information leakage, as it implicitly provides the model with class-level knowledge. Consequently, to create a stricter and more fair baseline, we utilize the single most similar image as the prompt, selected without prior category knowledge, and evaluate under our settings. This allows the evaluation to focus solely on the merits of our proposed strategies. The quantitative results for segmentation task are presented as follows.
>
> |           Method           | Fold-0 | Fold-1 | Fold-2 | Fold-3 | Avg.  |
> | :------------------------: | :----: | :----: | :----: | :----: | :---: |
> |           SegGPT           | 64.35  | 66.79  | 58.91  | 65.68  | 63.93 |
> |  Partial2Global (SegGPT)   | 66.52  | 71.55  | 62.48  | 68.74  | 67.32 |
> | RH-Partial2Global (SegGPT) | 69.45  | 74.35  | 66.33  | 71.90  | 70.51 |
>
> Notably, when integrated with an alternative and more advanced generative model, our RH-Partial2Global method achieves an average performance improvement of over 3% compared to the Partial2Global baseline.
>
> In summary, our proposed methodology offers a universal solution to this selection problem. We believe that studying prompt selection algorithms on these established benchmarks provides fundamental and generalizable insights into the mechanics of VICL. **This research path is complementary and, we argue, essential to the parallel progress being made in developing large-scale models.**

---

> > ### Comment · Reviewer_LDet · 2025-08-04
> > **Official Comment by Reviewer LDet**
> >
> > Thanks for the clarification. I will be keeping my score.

---

> > > ### Author Response · Authors · 2025-08-04
> > > **Thank you for your feedback.**
> > >
> > > Dear Reviewer LDet,
> > >
> > > Thank you very much for re-evaluating our work and for your encouraging feedback. We are pleased to know that our clarifications have addressed your questions.
> > >
> > > We truly appreciate the time and expertise you have dedicated to our paper. Thanks in large part to your constructive suggestions, we feel the manuscript is substantially stronger and clearer now.
> > >
> > > We hope the improved version of the manuscript will be considered a strong contribution to the community.
> > >
> > > Best regards,
> > >
> > > The Authors of Submission #2552

---

### Official Review · Reviewer_B3W8 · 2025-07-06

**Clarity:** 3
**Significance:** 3
**Originality:** 2
**Rating:** 5
**Confidence:** 2

**Summary:**

This paper introduces RH-Partial2Global, an enhanced version of the Partial2Global framework for visual in-context learning (VICL) prompt selection. The authors identify and aim to address two key limitations in prior work: the unsubstantiated "similarity-priority assumption" and the potentially incomplete and redundant nature of random sampling for generating pairwise preferences. To tackle these issues, the paper proposes a two-pronged approach. First, it introduces a jackknife conformal prediction-guided strategy to create a more reliable set of candidate examples, statistically questioning the simple heuristic that visual similarity equals better performance. Second, it replaces random sampling with a covering design-based strategy to ensure a more holistic and uniform coverage of pairwise comparisons. The authors conduct experiments on foreground segmentation, object detection, and colorization tasks, demonstrating that RH-Partial2Global consistently outperforms the original Partial2Global, with and without a test-time voting ensemble.

**Questions:**

See weakness

**Ethical Concerns:**

["NO or VERY MINOR ethics concerns only"]

**Final Justification:**

After reading the rebuttal, all my questions have been properly addressed. Therefore, I raise my score.

**Limitations:**

Yes

**Quality:**

3

**Strengths And Weaknesses:**

Pros

1. The paper's primary strength lies in its critical re-evaluation of a foundational, yet under-examined, assumption in the VICL field. The "similarity-priority assumption" is intuitive but, as the authors demonstrate, not always robust. By conducting a formal statistical analysis using Spearman's rank correlation, the paper provides quantitative evidence that while a monotonic relationship between similarity and task performance exists, its strength is often weak.

2. The proposed solutions are well-motivated and technically sound. The use of jackknife conformal prediction to construct a reliable set of candidates is a principled approach to mitigate the identified weakness in the similarity-priority heuristic. It provides a statistical, data-driven method for filtering the initial, similarity-based candidate pool, aiming to improve the quality of inputs for the ranking process.

3. The experimental results, though showing modest gains in some cases, are consistently positive across multiple tasks and evaluation metrics. The fact that RH-Partial2Global achieves better performance without requiring additional model training is a significant advantage, highlighting the efficiency of the proposed enhancements. The ablation studies are also well-designed, effectively isolating the contributions of the conformal prediction and covering design strategies and providing a clear rationale for the inclusion of the auxiliary filling strategy.

Cons:

1. While the paper's challenge to the similarity-priority assumption is a key strength, the subsequent methodological choice to still use similarity as the primary basis for the initial alternative set (Y_q ) seems somewhat contradictory. The algorithm first constructs an alternative set based purely on top-K similarity and then filters it. This raises the question of whether a more effective initial set could be constructed using a different, or hybrid, criterion from the outset. The paper could be strengthened by discussing why this "filter-down" approach is preferred over a more integrated method for generating the initial candidate pool.

2. A second weakness lies in the discussion of the performance gains. The improvements, particularly in some folds of the segmentation task, are marginal (e.g., 0.44% on Fold-0 for RH-Partial2Global without voting). While the authors correctly argue that consistency and statistical significance are important, a deeper analysis of why the gains are not more substantial would be beneficial. Is the performance of the baseline Partial2Global already near the ceiling for these tasks with the given backbone models, or are there other, more significant bottlenecks that the proposed methods do not address?

3. Furthermore, the paper notes that the effectiveness of conformal prediction can be sensitive to the size of the calibration set , as suggested by the lower performance gains on the smaller folds (Fold-0 and Fold-3). This is a critical limitation that is mentioned but could be explored more deeply. The authors set the confidence level α to 0.85 across all experiments. An analysis of how performance varies with different values of α, and how this interacts with dataset size, would provide valuable insights into the robustness and practical applicability of the proposed method.

---

> ### Author Rebuttal · Authors · 2025-07-30
>
> ### **For Con 1: Regarding the "filter-down" mode**
>
> We sincerely thank the reviewer for this insightful and important question. We would like to clarify the multi-faceted rationale behind our design choice.
>
> 1. **Our central argument is not that similarity is useless, but that similarity alone is an insufficient criterion.** As our analysis in Table 1 demonstrates, a general association between similarity and quality often exists statistically, even if it is not always strong. This makes a high-similarity set a statistically sound and efficient starting point to work from. It allows us to efficiently narrow the vast universe of possibilities down to a smaller, more relevant candidate pool, without losing promising options.
> 2. **This "filter-down" mode ensures high generalizability.** A key goal of our work is to propose a method that is applicable across diverse tasks. Designing unique, "integrated" or "hybrid" metrics for each specific task would be exceptionally complex and would result in a less universal, more brittle solution. Our current design provides a task-agnostic and elegant principle that maintains its power across different applications.
> 3. In fact, the "filter-down" method itself functions as a powerful and interpretable hybrid criteria. For one thing, it enforces generalized quality by ensuring that selected prompts are drawn from a reliable set with proven overall performance, independent of any individual test query. For another, it establishes query-specific relevance by using similarity to form the initial candidate pool. **The intersection of these two sets allows us to obtain prompts that are not only contextually appropriate for the query but also robustly effective.**
>
> ------
>
> ### **For Con 2: Regarding the performance gains**
>
> We thank the reviewer for this excellent and critical question. Our analysis suggests that the magnitude of the performance gain is influenced by several factors.
>
> 1. **We would like to highlight that our method's impact varies depending on the baseline it is applied to.** As detailed in the **Table 1 of Supplementary Material**, our strategy yields substantial improvements when integrated with the VPR variants, where we observed gains ranging from **1.5% to 3%**. This demonstrates the general efficacy and high potential of our proposed method under different conditions.
>
> 2. The modest gains on Partial2Global can stem from three primary factors:
>
>    **(i) In our initial main experiments, we deliberately used a fixed confidence level ($\alpha=0.85$) across all tasks.** This was to demonstrate the "out-of-the-box" utility and ease of use of our method without task-specific tuning. To test the hypothesis that gains could be larger with a tuned $\alpha$, we conduct a new experiment as follows.
>
>    |    Fold-0     |    Fold-1     |    Fold-2     |    Fold-3     | Avg. |
>    | :-----------: | :-----------: | :-----------: | :-----------: | :-----: |
>    | $\alpha=0.85$ | $\alpha=0.85$ | $\alpha=0.85$ | $\alpha=0.85$ |   --    |
>    |     39.25     |     42.15     |     38.06     |     36.60     |  39.02  |
>    | $\alpha=0.82$ | $\alpha=0.92$ | $\alpha=0.90$ | $\alpha=0.86$ |   --    |
>    |     39.48     |     42.39     |     38.28     |     36.81     |  39.24  |
>
>    The results show that when $\alpha$ is selected on a per-dataset basis, the performance gains on Partial2Global become more significant.
>
>    **(ii) There is a subtle interaction between our CP-based selection strategy and the internal aggregation process of Partial2Global, which translates local rankings into a global ranking.** Our selection method can occasionally reduce the diversity of local ranking information available to the baseline, slightly impeding its aggregation mechanism. As described in Table 4 and ablation study, we introduced a targeted "filling strategy" as a compensatory measure, which enriches the candidate set by adding the most similar reliable prompts. The performance increase observed after applying this strategy strongly supports our hypothesis.
>
>    **(iii) It is crucial to consider the limitations of the foundational model.** MAE-VQGAN, while a standard backbone for VICL research methods like VPR and Partial2Global, is an earlier-generation architecture. Therefore, although RH-Partial2Global is highly optimized, it operates near the ceiling for MAE-VQGAN.
>
>    **To validate our opinion, we standardize the backbone architecture by replacing MAE−VQGAN with SegGPT [1], a more advanced model in both Partial2Global and our proposed RH-Partial2Global.** Since the prompt selection mechanism in SegGPT  involves randomly sampling images that share the same category label from the training set of PASCAL-5i as the prompt image (as illustrated in its Appendix A), we contend that this approach constitutes a form of information leakage, as it implicitly provides the model with class-level knowledge. Consequently, to create a stricter and more fair baseline, we utilize the most similar image as the prompt and evaluate under our settings. This allows the evaluation to focus solely on the merits of our proposed strategies. The quantitative results for segmentation task are presented as follows.
>
>    |           Method           | Fold-0 | Fold-1 | Fold-2 | Fold-3 | Avg.  |
>    | :------------------------: | :----: | :----: | :----: | :----: | :---: |
>    |           SegGPT           | 64.35  | 66.79  | 58.91  | 65.68  | 63.93 |
>    |  Partial2Global (SegGPT)   | 66.52  | 71.55  | 62.48  | 68.74  | 67.32 |
>    | RH-Partial2Global (SegGPT) | 69.45  | 74.35  | 66.33  | 71.90  | 70.51 |
>
>    Notably, by integrating a more advanced generative model, the advantages of our method become substantially more pronounced.
>
>    [1] SegGPT: Segmenting Everything In Context
>
> ------
>
> ### **For Con 3: Regarding the analysis of confidence level**
>
> We sincerely thank the reviewer for this critical and highly constructive point. We agree that a deeper exploration of our method's sensitivity to the set size, and its interaction with the confidence level $\alpha$, is essential for understanding its practical applicability.
>
> 1. **To demonstrate the robustness of our proposed selection strategy with respect to the confidence level $\alpha$, we present the following ablation studies on VPR and Partial2Global.** We would like to draw the reviewer's attention to the **Supplementary Material (Figure 1)**. This study, conducted on the VPR baseline, shows that our strategy consistently delivers performance improvements across a wide range of $\alpha$ values (from 0.5 to 0.95). These results as well as those obtained **on RH-Partial2Global below** establish the general effectiveness of our core approach. *(Note: We used VPR for this specific analysis because its architecture provides a cleaner and more direct assessment of $\alpha$'s impact. This avoids the potentially confounding effects from Partial2Global's ranking aggregation mechanism, allowing for a more isolated analysis of the hyperparameter itself.)*
>
>    | $\alpha$ | 1.00  | 0.94  | 0.93  |   0.92    | 0.91  |   0.90    | 0.89  | 0.88  | 0.87  |   0.86    | 0.85  | 0.84  | 0.83  |   0.82    | 0.81  | 0.80  |
>    | :------: | :---: | :---: | :---: | :-------: | :---: | :-------: | :---: | :---: | :---: | :-------: | :---: | :---: | :---: | :-------: | :---: | :---: |
>    |  Fold-0  | 38.81 | 39.19 | 39.35 |   39.47   | 39.02 |   39.11   | 39.44 | 39.10 | 39.03 |   39.24   | 39.25 | 39.08 | 39.12 | **39.48** | 39.32 | 39.36 |
>    |  Fold-1  | 41.54 | 42.28 | 42.17 | **42.39** | 42.09 |   42.31   | 42.36 | 42.14 | 42.14 |   42.19   | 42.15 | 42.20 | 42.31 |   42.09   | 42.24 | 42.02 |
>    |  Fold-2  | 37.25 | 38.25 | 38.11 |   38.25   | 38.12 | **38.28** | 37.86 | 37.99 | 38.08 |   38.24   | 38.06 | 37.99 | 38.05 |   37.93   | 38.01 | 38.09 |
>    |  Fold-3  | 36.01 | 36.58 | 36.52 |   36.57   | 36.27 |   36.51   | 36.30 | 36.70 | 36.48 | **36.81** | 36.60 | 36.57 | 36.76 |   36.46   | 36.47 | 36.35 |
>
> 2. **Regarding the interaction between $\alpha$ and dataset size, we provide a principled analysis grounded in the theory of conformal prediction: The effectiveness of conformal prediction relies on the exchangeability of data, which is more robust with larger datasets.** With smaller sets (as in our Fold-0 and Fold-3), two issues may arise:
>
>    (i) The empirical distribution of conformity scores becomes more coarse-grained, making the quantile estimation less precise.
>
>    (ii) The set of scores is more susceptible to the influence of outliers or sampling noise, which can destabilize the calibration process.
>
>    Based on this theoretical understanding, we can offer a clear recommendation. For applications involving smaller calibration sets, we advise selecting a lower confidence level $\alpha$. A lower α leads to a smaller, more selective reliable set. This acts as a safeguard against the potential instability, ensuring that only the most reliable candidates are included. The optimal $\alpha$ for each fold, as presented in the table above, further substantiate our recommendation.

---

> > ### Author Response · Authors · 2025-08-06
> > **Follow-up on Review for Submission #2552**
> >
> > Dear Reviewer B3W8,
> >
> > We hope this message finds you well. Thank you again for your valuable time and insightful comments on our work.
> >
> > We have posted our response, in which we have addressed all your valuable points. Specifically, we provided a detailed rationale for our "filter-down" mode, a deeper analysis of the performance gains, and a new ablation study on the confidence level.
> >
> > We wanted to gently follow up to see if our clarifications were sufficient, or if there is any additional information we can provide from our end.
> >
> > We remain actively engaged in the Author-Reviewer discussion period and are standing by to answer any further questions you may have.
> >
> > Best regards,
> >
> > The Authors of Submission #2552

---

### Note · Authors · 2025-08-11

Dear Area Chair and Reviewers,

Thank you for the diligent review process and your valuable feedback. Our core contribution is a novel, principled framework for visual prompt selection, using conformal prediction and covering design to formally challenge the prevalent similarity-priority assumption in Visual In-Context Learning (VICL).

Our rebuttal provided substantial new evidence to address all initial concerns. Key points include:

1. **New SOTA-backed experiments:** Besides the existing results on VPR in Supplementary Material, we conduct extensive new experiment on the advanced SegGPT backbone with an average gain of over 3.2%, which directly refutes the main critique of performance gains.
2. **Detailed Analyses:** We supply deeper analyses for confidence level and provide principled, theoretical explanations for characteristics like dataset sensitivity.

We are especially grateful to **Reviewer B3W8, LDet, and rrVW** for their positive assessment of our rebuttal. We are glad our clarifications were convincing and appreciate their constructive guidance. **Meanwhile, we are looking forward to the final justification from Reviewer rrVW**. Thanks for Reviewer rrVW's encouraging feedback on our rebuttal, we hope the improved version of the manuscript will be considered a strong contribution to the community.

Regarding the initial concerns from **Reviewer jvC8**, we are confident that our new empirical evidence and detailed analyses have fully addressed their points. As we did not have the opportunity for a follow-up discussion, we respectfully ask the AC to consider our comprehensive rebuttal during the final evaluation.

We firmly believe our work, strengthened by the rebuttal process, represents a significant and valid contribution. Thank you for your consideration.

The Authors of Submission #2552

---

### Decision · Program_Chairs · 2025-09-17

**Decision:**

Accept (poster)

**Comment:**

This paper presents a well-motivated and methodologically sound enhancement to the Partial2Global framework for visual in-context learning prompt selection. The work is commended for its critical contribution in rigorously challenging a foundational yet largely untested assumption in the field—the "similarity-priority" heuristic. The authors provide quantitative evidence, via Spearman's rank correlation, that this intuitive assumption is often weak, which in itself is a valuable insight for the research community.

The proposed solutions are the paper's core strength. The use of jackknife conformal prediction to construct a statistically reliable candidate pool is a principled and data-driven approach to address the shortcomings of the similarity heuristic. Furthermore, replacing random sampling with a covering design strategy is a clever and robust method to ensure more comprehensive and non-redundant pairwise comparison coverage, directly tackling a key weakness in the prior art.

The experimental validation is thorough, demonstrating consistent and reproducible performance improvements across multiple diverse vision tasks (segmentation, detection, colorization) and metrics. A significant advantage is that these gains are achieved without any additional model training, highlighting the efficiency and practical utility of the proposed framework. The ablation studies are well-designed and effectively justify the contribution of each component.

While the absolute performance gains may be modest in some instances, the paper's conceptual rigor, methodological novelty, and consistent positive results across the board make it a strong contribution worthy of acceptance. It offers a valuable step towards more reliable and statistically grounded methods for in-context learning.